# High-Frequency and Spectrum-Clean Shear-Horizontal Acoustic Wave Resonators with AlN Overlay

**DOI:** 10.3390/mi13071029

**Published:** 2022-06-29

**Authors:** Zonglin Wu, Shuxian Wu, Feihong Bao, Jie Zou

**Affiliations:** The Key Laboratory for Information Science of Electromagnetic Waves, School of Information Science and Technology, Fudan University, Shanghai 200438, China; 20210720006@fudan.edu.cn (Z.W.); sxwu20@fudan.edu.cn (S.W.); baofh@fudan.edu.cn (F.B.)

**Keywords:** aluminum nitride, LiNbO_3_, 6G, shear-horizontal mode, coupling coefficient

## Abstract

By bonding the sub-wavelength-thick lithium niobate (LiNbO_3_) layer to high-phase-velocity (*v*_p_) substrates, such as Si, the shear-horizontal (SH) modes no longer couple with the bulk modes leaking into substrates. As the propagation loss is no longer the major concern for these types of nonleaky SH wave devices, the *YX*-LiNbO_3_ with a low rotation angle providing ultra-large coupling coefficient (*k*_eff_^2^) can be used. In addition, by overlaying a high-velocity layer such as AlN on top of LiNbO_3_/Si, the *v*_p_ of the SH wave can be significantly enhanced at a small cost of *k*_eff_^2^. By a careful design of the stack, both the wide-band spurious (Lamb wave) and near-band spurious (Rayleigh wave) are suppressed successfully. This paper focuses on the design of layered substrate not only to optimize its resonance characteristics—series frequency (*f*_s_*)*, quality factor (*Q)*, *k*_eff_^2^, and temperature coefficient of frequency (*TCF*)—but also for eliminating the out-of-band spurious responses. The optimized substrate design demonstrates the minimal propagation loss, high *f*_s_ of 3 GHz, large *k*_eff_^2^ of 14.4% and a spurious-free response at 0–6 GHz. These novel nonleaky SH wave devices can potentially enable the low loss and wideband processing functions, which is promising for the 5G/6G radio frequency (RF) communication systems.

## 1. Introduction

The propagation-loss mechanisms of shear-horizontal (SH)-type waves on LiNbO_3_ and LiTaO_3_ have been intensively studied recently [1,2,3,4,5,6]. By bonding the sub-wavelength-thick LiNbO_3_ or LiTaO_3_ layer to a high-velocity substrate, the SH mode no longer couples with the bulk mode in substrate and the leaky component is effectively eliminated [1,2,3,4,5,6]; thus, the SH wave can abandon its traditional name—“leaky-surface acoustic wave (SAW)” but a favorable “nonleaky” feature instead. There are many choices of high-velocity substrates, such as AlN, Si, quartz, diamond, etc. [1,2,3,4,5], and Si is used as an example in this study for its wide availability and cheap price. In addition, given that the acoustic propagation loss no longer dictates the choice of piezoelectric-substrate cut angle and ultra-large effective coupling coefficient (*k*_eff_^2^), cuts can be taken advantage of, such as low rotation from *YX*-LiNbO_3_ [3,4,5,6,7].

Long-Term Evolution (LTE)—Advancement Pro, 5G sub-6 GHz new radio (NR), and emerging 6G standards require low-loss and wide-bandwidth (fractional bandwidth > 5%) filters that push the frequency limits SAW technology using optical lithography up to more than 2.5 GHz range [8,9,10,11,12,13,14]. That shrinking SAW resonators size enables the high frequency, while the quality-factor (*Q*) significantly deteriorates. With the *k*_eff_^2^ less than 10% and phase velocity (*v*_p_) less than 4500 m/s, the current commercially popular standard-SAW and Temperature Compensated Surface Acoustic Wave (TCSAW) devices can hardly meet the requirements. Intriguingly, the low-angle-rotated *YX*-LiNbO_3_ provides ultralarge intrinsic *k*_eff_^2^ (~30%). After LiNbO_3_ bonded to Si, the propagation-loss problem is resolved and high *Q* is achieved [4]. The multilayered LiNbO_3_ SAW devices with ultralarge *k*_eff_^2^ and high *Q* meet requirements in low frequency (<2.5 GHz)_._ However, the *v*_p_ is still relatively low (~4000 m/s), limiting its application in LTE high-band (2.5–3.5 GHz, such as band 7, band41, etc.) filters and 5G NR applications. A high-velocity layer beneath the piezoelectric LiNbO_3_ or LiTaO_3_ does not effectively increase the phase velocity of the SH mode; rather, by adding a high-velocity layer, such as AlN, overlaid on top of the transducer and LiNbO_3_, the phase velocity of the SH mode is visibly increased with only minimal trade-off on the *k*_eff_^2^, which can be much larger than needed for majority of the bands. In this way, a desirable nonleaky SH wave resonator with low propagation loss, high frequency (*f*), and large-*k*_eff_^2^ can be achieved.

In addition, without a careful design of the substrate, the layered SH potentially has spurious responses in the out-of-band frequencies (Lamb modes) and near-band frequencies. This paper focuses on the design of layered substrate not only to optimize its narrow-band characteristics—*f*, *Q*, *k*_eff_^2^, temperature coefficient of frequency (TCF)—but also for eliminating the out-of-band spurious responses.

By using the numerical calculation and finite-element-method (FEM) approaches, propagation characteristics of the SH modes in LiNbO_3_/Si and AlN/LiNbO_3_/Si are derived with varied substrate designs and the device performance is optimized as well as the presence of spurious modes avoided. Based on the analyses, new stacks with optimized substrate design are obtained, demonstrating the high frequency of 3 GHz at 1 µm interdigital transducer (IDT) pitch, high *k*_eff_^2^ (14.4%) and a spurious free response at 0–6 GHz. After optimizations, these novel AlN/LiNbO_3_/Si nonleaky SH devices, featuring superb performance with high frequency ability and being spurious-free, show great potential for filters in next-generation RF front ends.

Symbols used in this paper are listed in Table 1.

## 2. Substrate Leakage

Figure 1 depicts the FEM-simulated mode shapes of the SH wave propagating in LiNbO_3_ substrate, LiNbO_3_/Si layered substrate, and AlN/LiNbO_3_/Si layered substrate. Perfect-matched-layer (PML) physics is assigned to bottom layers of the 3D unit cell models to simulate the substrate that is too thick, comparing it to wavelengths to generate wave reflections from its bottom. The periodic conditions are applied to both *x* (perpendicular to IDT fingers) and *y* (in parallel with IDT fingers) directions so that the basic semi-infinite plane condition is assumed for the wave propagation. The material constants are from [14,15,16,17] and are listed in Table 2. In the structures shown, *h*_LiNbO3_/*λ* = 0.25, *h*_AlN_/*λ* = 0.2 are assumed; 8%*λ*-thick aluminum (Al) is used as IDT electrodes here and throughout the paper.

As shown in Figure 1a, the shear component in *YX*-LiNbO_3_ substrate still concentrates at the surface, but strong leakage results from coupling to the slow shear bulk acoustic wave and mechanically propagates down obliquely, as shown in *u_z_* and *u_x_*. The lower-case coordinate *x* is the propagation direction proportional to IDT, *y* is the transversal direction parallel to IDT, and *z* the direction down into substrate. By bonding the high-velocity Si substrate to the sub-wavelength *YX*-LiNbO_3_, the SH wave in LiNbO_3_/Si (Figure 1b) no longer couples with the bulk mode and leaks into the substrate since the bulk velocity is higher than the SH mode on surface. After adding an AlN coating layer, the SH mode still concentrates in the LiNbO_3_ piezoelectric layer and minimizes the leaky component into the Si substrate (Figure 1c).

## 3. Propagation Characteristics of SH Wave in LiNbO_3_/Si

The *v*_p_ and *k*_eff_^2^ of the SH waves propagating in the single and layered piezoelectric substrates can be theoretically calculated using numerical analysis—Adler’s matrix approach [17] or FEM [18]. The material constants for both the calculation and FEM simulation are also listed in Table 2. The numerically calculated propagation characteristics are checked with FEM simulation and agreed well with the FEM results. In the FEM simulation, the model is a 3D building-block cell with periodic boundary conditions (BCs) on both *x*- and *y*- directions. All the *k*_eff_^2^ are derived from series-resonance frequency (*f*_s_) and parallel-resonance frequency (*f*_p_) using the *IEEE* standard definition of the device electromechanical coupling [19]:(1)$$k_{\mathrm{eff}}^{2}=\frac{\pi }{2}\frac{f_{s}}{f_{p}}{\left[\tan(\frac{\pi }{2}\frac{f_{s}}{f_{p}})\right]}^{-1}.$$

Comparing to the intrinsic coupling coefficient (*k*
_intr_^2^) derived from the difference from the open- and metallized-surface *v*_p_:(2)$$k_{\mathrm{intr}}^{2}=\frac{v_{p,o}^{2}-v_{p,m}^{2}}{v_{p,o}^{2}}$$
the device-level *k*_eff_^2^ yields close values to the *k*_intr_^2^ and considers the electric field more accurately. The differences of these two definitions of couplings are discussed and compared in detail in Figure 2 of the reference [18]_._ Despite the similarity for a standard device, the impact of the actual electrode cross-sectional shape and coverage ratio can be considered in the *k*_eff_^2^ derivation but not *k*_intr_^2^, so the *k*_eff_^2^ evaluation offers better accuracy and potential for the future device optimization. As a result, the device-level coupling coefficient *k*_eff_^2^ is utilized throughout the analysis of this work.

### 3.1. Cut Angle

As the propagation loss is no longer the main concern and dictates the cut-angle selection for the nonleaky SH wave in bonded wafers, the optimal cut angle can be chosen to optimize *v*_p_ and *k*_eff_^2^. Figure 2a,b depicts the open-surface phase velocities *v*_p,o_ and *k*_eff_^2^ for LiNbO_3_/Si across all rotation angles *θ* from the *YX*-LiNbO_3_ with varied thicknesses of LiNbO_3_ for both the SH main mode and the Rayleigh spurious mode.

The impact of the rotation angle on the propagation characteristics of SH mode in LiNbO_3_/Si layered substrate is similar to on the LiNbO_3_ substrate [20], as depicted in Figure 2a. Fortunately, a low rotation angle from *YX*-LiNbO_3_ enables high *v*_p,o_, large *k*_eff_^2^ for the SH main mode, and low *k*_eff_^2^ for Rayleigh mode simultaneously. Therefore, the design range of *θ* is 10–40° for the LiNbO_3_/Si-based nonleaky SH wave devices for the high frequency, wide band, and clean spectrum, respectively.

In Figure 2b, it can also be noted that *h*_LiNbO3_ = 0.25*λ* enables higher *v*_p,o_ than *h*_LiNbO3_ = 0.5*λ* of LiNbO_3_/Si or LiNbO_3_ substrate, and the optimized rotation angle also shifts up a bit for the optimum *k*_eff_^2^. These indicate weak dispersion in the layered substrate due to the sub-wavelength-thick piezo thin film.

### 3.2. Dispersion and Spurious Modes

As shown in Figure 3a,b, the dispersive curves of the *v*_p,o_’s and *k*_eff_^2′^s of the SH main mode, Rayleigh spurious mode, and S_0_ Lamb mode spurious mode propagating in the LiNbO_3_/Si bonded structure with varied rotation angle of *YX*-LiNbO_3_ are presented. Due to the sub-wavelength thick piezo-layer structure, the SH mode and Rayleigh mode show weak dispersive characteristics in the phase velocities. The S_0_ Lamb wave, however, shows even stronger dispersion due to its plate-wave type whose characteristics are usually impacted a lot by the piezoelectric-layer thickness normalized to wavelength. Although varying LiNbO_3_ thickness does not move the Rayleigh spurious mode away from the SH main mode in frequency, engineering the LiNbO_3_ thickness does effectively keep the Lamb modes distant in frequency from the SH mode. Luckily, when the *h*_LiNbO3_ < 0.5*λ*, the closest Lamb mode S_0_ mode would be 20% higher in frequency than the SH main mode, which makes a distance of at least 500 MHz if the SH passband is at 2.5 GHz.

The dispersive characteristics in *k*_eff_^2^ is stronger than in *v*_p,o_ for the SH mode, as shown in Figure 3b. In other words, the change of *k*_eff_^2^ for SH mode with *h*_LiNbO3_ is more obvious than that of *v*_p,o._ In order to achieve a high *k*_eff_^2^, the LiNbO_3_ cannot be too thin or too thick and the design range is preferred to be > 0.2*λ* and < 0.5*λ*. In addition, the Rayleigh and S_0_ Lamb spurious modes can also be suppressed by choosing the LiNbO_3_ thickness and cut-angle combination smartly. For the S_0_ Lamb mode, the case is simpler since its *k*_eff_^2^ will lower at smaller LiNbO_3_ thicknesses under different LiNbO_3_ cut angle, and the design range of *h*_LiNbO3_ < 0.5*λ* fortunately happens to be able to suppress the S_0_ Lamb modes. As a contrary, the LiNbO_3_ thickness and rotation angle have to be optimized together for the Rayleigh mode, and a slightly larger rotation angle is preferred for the 0.2*λ* < *h*_LiNbO3_ < 0.5*λ* design range, such as 30° *YX*-LiNbO_3_.

### 3.3. Frequency Response

Figure 4a,b show the FEM-simulated narrow-band response compared to LiNbO_3_ substrate and wide-band response with periodic structure, which is a 1.5-dimension (1.5 D) model based on LiNbO_3_/Si with 30° *YX*-LiNbO_3_ and *h*_LiNbO3_*/λ* = 0.25. By the 1.5D model we assume an infinite number of IDT fingers (*NF*) and infinite aperture lengths, but the stack setup in *z* direction is fully considered (1D), as well as the IDT duty factor (*DF = finger width/pitch*) and IDT shape in periodicity (0.5D).

Comparing the conductance and admittance curves of the SH modes in LiNbO_3_ substrate and LiNbO_3_/Si in Figure 4a, it is clearly seen that in LiNbO_3_ substrate, as the bulk wave velocity is lower than SH wave, the antiresonance is distorted with low-*Q*, and from the conductance curve it could be observed that the wave cannot be effectively reflected in the stopband as well. On the contrary, for LiNbO_3_/Si stack, the parallel resonance features a very sharp response and the conductance level is very deep around the *f*_p_ and throughout the stopband thanks to the minimal bulk radiation, again indicating the ultra-low propagation loss and the fact of nonleaky characteristics.

In addition, for the LiNbO_3_/Si and at this LiNbO_3_ cut angle and LiNbO_3_ layer thickness designed to lower the Rayleigh *k*_eff_^2^ to near-zero, the narrow-band response (Figure 4a) is clean from the Rayleigh mode, which presents in the response based on LiNbO_3_ substrate. On the other hand, since the LiNbO_3_ is thin enough, the wide-band response of the LiNbO_3_/Si resonator also shows an extremely clean spectrum from 0 to 6 GHz.

## 4. Propagation Characteristics of SH Wave in AlN/LiNbO_3_/Si

### 4.1. Cut Angle

Figure 5a,b depict the open-surface phase velocities *v*_p,o_ for varied normalized thicknesses of AlN overlay layer on top of the IDT transducers sitting on LiNbO_3_/Si and the piezo-layer is across all rotation angles from the *YX*-LiNbO_3_ with *h*_LiNbO3_*/λ* = 0.25 and *h*_LiNbO3_*/λ* = 0.5, respectively. The *c*-axis-oriented AlN material constants are from literature [15] and listed in Table 2. The nonleaky SH wave propagating in the AlN/LiNbO_3_/Si stack shows a similar trend to that in the LiNbO_3_/Si layered stack. For both LiNbO_3_ thickness cases, the phase velocity can be effectively enhanced by the AlN overlay, and the improvement converges when *h*_AlN_ is larger than 0.2*λ*. The phase velocities of the SH modes in Figure 5a with *h*_LiNbO3_*/λ* = 0.25 are in general larger than the case in Figure 5b with *h*_LiNbO3_*/λ* = 0.5. As can be observed in Figure 5a, with *h*_LiNbO3_*/λ* = 0.25 and for *θ* between 20° and 80°, the *v*_p,o_ can be as high as above 5000 m/s thanks to the AlN coating with *h*_AlN_ > 0.2*λ*.

Furthermore, the Rayleigh mode in the case of *h*_LiNbO3_*/λ* = 0.25 are less coupled with the SH mode at a high rotation angle of around 128°, and also less perturbed by *θ* than the case of *h*_LiNbO3_*/λ* = 0.5. At low *θ* < 30°; however, the Rayleigh spurious mode is slightly closer to the SH mode in the case of *h*_LiNbO3_*/λ* = 0.25 than in the thicker case, which is in a similar trend with Figure 3a.

Figure 6a and b depict the effective coupling coefficient *k*_eff_^2^ for varied thicknesses of AlN overlay layer on LiNbO_3_/Si across all rotation angles from the *YX* LiNbO_3_ with *h*_LiNbO3_*/λ* = 0.25 and *h*_LiNbO3_*/λ* = 0.5, respectively. Although the AlN overlay apparently lowers the *k*_eff_^2^, the degraded *k*_eff_^2^ would still be more than enough and much larger than the current technologies. Moreover, the reduction in *k*_eff_^2^ would converge when *h*_AlN_ is larger than 0.2*λ*. For both LiNbO_3_ thickness cases, and for cut angles between 0° and 60°, the *k*_eff_^2^ can be as high as above 11%. Comparing two LiNbO_3_ thicknesses, the peak *k*_eff_^2^ of the SH mode in AlN/LiNbO_3_/Si across a wide rotation-angle range are at similar level.

It is also interesting to note that at *θ* ~ 10°–30°, the *k*_eff_^2^ of SH mode is maximized and at the same time the *k*_eff_^2^ of Rayleigh mode is minimized, where the Rayleigh spurious mode can be suppressed in the nonleaky SH SAW resonator or filter. With AlN overlays, the optimized cut angle for Rayleigh mode elimination shifts down a bit. The optimal cut-angle design range would be of 10°–30° for simultaneously achieving high *v*_p,o_ and large *k*_eff_^2^ for the SH mode, as well as low- *k*_eff_^2^ for Rayleigh mode.

### 4.2. Trade-Offs between v_p,o_ and k_eff_^2^

The trade-offs between the *v*_p,o_ and *k*_eff_^2^ by varying AlN thickness are compared for the nonleaky SH wave propagating in AlN/LiNbO_3_/Si with two different rotation angles of the piezoelectric LiNbO_3_, as presented in Figure 7a,b. It can be noted that both the *v*_p,o_ and *k*_eff_^2^ saturate when *h*_AlN_ > 0.4*λ*. For both 15° *YX*-LiNbO_3_ and 30° *YX*-LiNbO_3_, the *v*_p,o_ is much higher for the case of *h*_LiNbO3_ = 0.25*λ* than *h*_LiNbO3_ = 0.5*λ*, and the *k*_eff_^2^ is also slightly higher for the thinner case. At *θ* = 15°, the saturated *v*_p,o_ is as high as 5280 m/s when *h*_LiNbO3_ = 0.25*λ*; at *θ* = 30°, the saturated *v*_p,o_ is 5240 m/s when *h*_LiNbO3_ = 0.25*λ*. The preferred AlN thickness design range would be between 0.2*λ* and 0.4*λ* right before the convergence in order to avoid additional mass loading on the device coupling.

Figure 8 depicts the displacement field as well as the first principal stress field of the nonleaky SH mode on the AlN/LiNbO_3_/Si with increasing AlN normalized thicknesses. It is most obvious that the mechanical fields become more penetrated and uniform when AlN becomes thicker. It is also intriguing to note that when the AlN layer is thicker than 0.4*λ*, the vibration becomes off the surface and concentrated in the highly piezo LiNbO_3_ layer; the AlN film then starts to be free of the mechanical vibration and transduction, indicating a stable mechanical-loading effect only instead of wave perturbation.

Both the saturated values of *v*_p,o_ and *k*_eff_^2^ are larger in the case of *h*_LiNbO3_ = 0.25*λ* compared to *h*_LiNbO3_ = 0.5*λ* for either 15° *YX*-LiNbO_3_ or 30° *YX*-LiNbO_3_. Note at for the 15° *YX*-LiNbO_3_ case, 0.5*λ*-thick LiNbO_3_ yields larger *k*_eff_^2^ than 0.25*λ*-thick LiNbO_3_ when AlN overlay is not applied and *h*_AlN_ = 0, which can also be observed from Figure 4b. However, even with slight AlN overlay, the *k*_eff_^2^ of 0.25*λ*-thick LiNbO_3_ becomes similar or larger. As a result, for both rotation-angle cases, the 0.25*λ*-thick LiNbO_3_ enables much higher *v*_p,o_ and similar *k*_eff_^2^.

When *h*_LiNbO3_ = 0.25*λ*, the *v*_p,o_ can be effectively boosted from 4420 m/s to 5280 m/s, showing a near 20% increase when *h*_AlN_ is up to > 0.4*λ*. Although the *k*_eff_^2^ is decreased by increasing *h*_AlN_, the absolute value is still above 14% even with a large *h*_AlN_, which is sufficient for most commercial bandwidths, thanks to the super-large intrinsic material electromechanical coupling *K*^2^ of the low-angle-rotated *YX*-LiNbO_3_.

### 4.3. Rayleigh Spurious

With the ability of high electromechanical coupling, the Rayleigh mode performs as the major spurious mode for most SH main-mode devices. In addition to the phase velocities of the Rayleigh mode always being very close to the SH mode, the Rayleigh spurious mode could generate prominent passband notches and near-band spikes for the SH wave filters, and pose severe risks for the application of the nonleaky SH waves. Therefore, the suppression of the Rayleigh spurious mode is highly desirable.

Figure 9 depicts the simulated *k*_eff_^2^ of the Rayleigh spurious mode versus AlN thicknesses for the AlN/LiNbO_3_/Si with different rotation angle and *h*_LiNbO3_*/λ* = 0.25. While the 30° and 35° rotation angles enable near-zero *k*_eff_^2^ of the Rayleigh mode, when AlN overlay becomes thicker, the preferred rotation angle is smaller for the low *k*_eff_^2^ of the Rayleigh mode. Or, in other words, for different rotation angles of the LiNbO_3_ layer, the optimized AlN thicknesses for zero-coupling Rayleigh mode are varied: for relatively lower rotation angle, the optimized AlN thickness would be large to diminish the Rayleigh mode.

As concluded in the previous section, 0.2*λ*–0.4*λ* thick AlN overlay is preferred for enabling the large velocity and *k*_eff_^2^ level at the same time. The optimized cut angle for the near-zero coupling of the Rayleigh spurious mode would be between 10° and 15° *YX*-LiNbO_3_, as shown in the yellow and green curves as examples inside the design range marked in Figure 9. Again, from the green and blue curves in Figure 6a, it can be found that 15° *YX*-LiNbO_3_ with 0.2*λ* AlN enables larger *k*_eff_^2^ of the main mode than 10° *YX*-LiNbO_3_ with 0.3*λ* AlN. As a result, 15° *YX*-LiNbO_3_ with 0.2*λ* AlN can be chosen for a high suppression of the Rayleigh spurious mode, as well as enabling large *v*_p_ and high *k*_eff_^2^ for the main SH main mode simultaneously.

### 4.4. Improvement of TCF

In addition, without a careful design of the substrate, the layered SH potentially has spurious responses in the out-of-band frequencies (Lamb modes) and near-band frequencies. This paper focuses on the design of layered substrate not only to optimize its narrow-band characteristics—*f*, *Q*, *k*_eff_^2^, temperature coefficient of frequency (TCF)—but also for eliminating the out-of-band spurious responses.

The *TCF* performance measuring the thermal stability of a resonator is set by the temperature dependence of phase velocity and the thermal-expansion coefficient of the wave along the propagation direction. The first-order *TCF*’s for the series resonance (*TCF*_s,1st_) and parallel resonance (*TCF*_p,1st_) are calculated as
(3)$$TCF_{s,1st}=\frac{1}{f_{s}}\frac{\partial f_{s}}{T}=\frac{1}{v_{p,SC}}\frac{\partial v_{p,SC}}{T}-\alpha_{x},$$
(4)$$TCF_{p,1st}=\frac{1}{f_{p}}\frac{\partial f_{p}}{T}=\frac{1}{v_{p,OC}}\frac{\partial v_{p,OC}}{T}-\alpha_{x},$$
where *v*_p,SC_ and *v*_p,OC_ refer to the phase velocities under short-circuited (SC) and open-circuited (OC) grating BCs, shown in the inset of FIG 10. From the coupling-of-modes (COM) theory, these BCs corresponds to the *f*_s_ and *f*_p_, respectively. Their temperature dependence ∂/∂*T* is calculated from the temperature coefficients of stiffness constants, temperature coefficients of piezoelectric constants, and temperature coefficients of permittivity of LiNbO_3_, AlN, Si, and Al listed in Table 3. The *α_x_* corresponds to the thermal-expansion coefficient of the substrate in the wave-propagation direction *x*, also listed in Table 3. Since the Si substrate is much thicker than the LiNbO_3_ and AlN, the effect of thermal expansion is limited by the clamping substrate Si, and its *α*_11_ of 2.6 ppm/°C from literature [19] is used herein for the derivation. 

Figure 10 shows the calculated *TCF*_s,1st_ and *TCF*_p,1st_ by varying the AlN thickness for the nonleaky SH waves propagating in AlN/LiNbO_3_/Si with the 0.25*λ*- thick 15° *YX*-LiNbO_3_ optimized from the previous analysis. Although AlN also becomes softer (contributing to *TCV*) and larger (contributing to *α*) when temperature rises, its *TCF* absolute value is much lower than LiNbO_3_—~−26 ppm/°C. Thus, the thicker AlN can reduce the thermal dependence of the phase velocity for the SH wave traveling in the composite structure. The *TCF*_p,1st_ is always lower than *TCF*_s,1st_ due to the positive *Te*_15_ and *Te*_22_ of LiNbO_3_; at a high temperature, *k*_eff_^2^ would increase slightly.

### 4.5. Slowness Curve and Propagation Direction

Although the propagation direction can be lithographically controlled in most cases along the *X* direction (all the previous analysis assumes the *X* propagation direction), the polar plots of propagating characteristics versus propagation direction can be good indicators for understanding the wave properties as well as fostering the device design [20]. Figure 11a,b show the slowness (*S*) curve and the *k*_eff_^2^ of the SH wave versus different propagation directions on the LiNbO_3_ substrate, LiNbO_3_/Si (*h*_LiNbO3_*/λ* = 0.25), and AlN/LiNbO_3_/Si (*h*_LiNbO3_*/λ* = 0.25, *h*_AlN_*/λ* = 0.2). The slowness curve for the LiNbO_3_ substrate shows a concave feature, whereas for the LiNbO_3_/Si and AlN/ LiNbO_3_/Si stacks it is nearly straight near the *x*-axis, indicating minimal diffraction of the nonleaky SH wave. Usually, for a well-guided wave with convex slowness curves, a faster region is required at the lateral ends to guide the wave, and for the concave case there might be lateral leakage to the fast regions. For the “straight” type [3], the IDT gap region design would be different and less sensitive to the concave or convex cases. In addition, from Figure 11a, the phase *v*_p_’s is largely enhanced in all propagation directions after adding the AlN overlay.

In Figure 11b, the *k*_eff_^2^ decreases drastically when the propagation direction deviates from the *X*-axis. In propagation directions close to *X*, the *k*_eff_^2^ is slightly reduced from LiNbO_3_ substrate to the LiNbO_3_/Si nonleaky stack (agreeing with Figure 2b in the X axis case); in propagation directions close to *Z*, the *k*_eff_^2^ is very slightly improved from LiNbO_3_ substrate to the LiNbO_3_/Si bonded structure. After adding the AlN overlay on top of the transducer, the *k*_eff_^2^ reduces drastically due to the mechanical loading effect, and in the *X* direction the *k*_eff_^2^ falls to between 10% and 15%, which is still more than enough for the advanced LTE bandwidth specification and most of the 5G NR bands.

In summary, the optimized propagation direction for the AlN/LiNbO_3_/Si resonator is the material *X* direction of LiNbO_3_ thanks for the fast wave-travelling velocity and the ultralarge *k*_eff_^2^. The wave is also better-guided in the transversal direction compared to the traditional leaky-SH resonator based on LiNbO_3_ substrate.

### 4.6. Frequency Response

Combining the previous analysis toward a high-*f*, large- *k*_eff_^2^, and spurious-free response utilizing the nonleaky SH wave, a stack with optimized substrate values is achieved: 5° *YX*-LiNbO_3_, *h*_LiNbO3_*/λ* = 0.25, and *h*_AlN_*/λ* = 0.2. Figure 12a,b plot the FEM-simulated narrow-band responses of the SH wave in the AlN/LiNbO_3_/Si layered stack compared to LiNbO_3_/Si and LiNbO_3_ substrate, as well as wide-band response with periodic structure (1.5D model). Intriguingly, the AlN/LiNbO_3_/Si layered structure enables frequency as high as 3 GHz while the IDT pitch is as large as 1 μm, ensuing good power handling. The *f* can further scale up if smaller *λ* is employed. The frequency or *v*_p_ has been increased by 18%, breaking the frequency limits for SAW resonators and filters and applicable to high-frequency bands in LTE-Advancement Pro and 5G NR.

Comparing admittance curves of the SH modes in AlN/ LiNbO_3_/Si, LiNbO_3_/Si, and LiNbO_3_ substrate in Figure 12a, it can be noted that the antiresonances for both AlN/LiNbO_3_/Si and LiNbO_3_/Si are very sharp, indicating a very high-quality factor at parallel-resonance (*Q*_p_) due to the elimination of bulk leakage (quality factor at series resonance (*Q*_s_) is usually dominated by the transducer resistance *R*_s_ and *Q*_p_ dominated by the acoustic propagation loss).

In addition, the AlN/ LiNbO_3_/Si layered substrate with the optimized parameters shows an extremely clean response in both narrow and wide spectrums. The Rayleigh mode and Lamb modes are suppressed with *k*_eff_^2^ of near-zero. By the analysis and careful design, the novel stack provides high performance, the ability of high frequency, and an extremely clean spectrum from 0 to 6 GHz simultaneously.

## 5. Conclusions

In this study, high-frequency nonleaky SH SAW resonators on AlN/LiNbO_3_/Si are demonstrated with thermal stability, large coupling and spurious-free. The high-velocity Si substrate was used to reduce the propagation leakage into substrate, and 15–30° rotation angles from *YX*-LiNbO_3_ were selected to provide the ultralarge *k*_eff_^2^. A careful trade-off analysis is provided on the AlN coating thickness between the enhanced *v*_p_ (*f* for given transducer) and the cost of extra *k*_eff_^2^. Furthermore, the out-of-band and near-band spurious modes in the layered SAW structures were analyzed by using FEM simulations and the requirements for the substrate were derived to avoid the presence of spurious modes. Based on this analysis, a new nonleaky SH resonators with optimized substrate design are obtained, demonstrating the ability with a high frequency of 3 GHz at 1 μm IDT pitch, a high *k*_eff_^2^ of 14.4% and a spurious-free response throughout 0–6 GHz, showing a great potential for 5G/6G RF communication systems.

## Figures and Tables

**Figure 1 micromachines-13-01029-f001:**
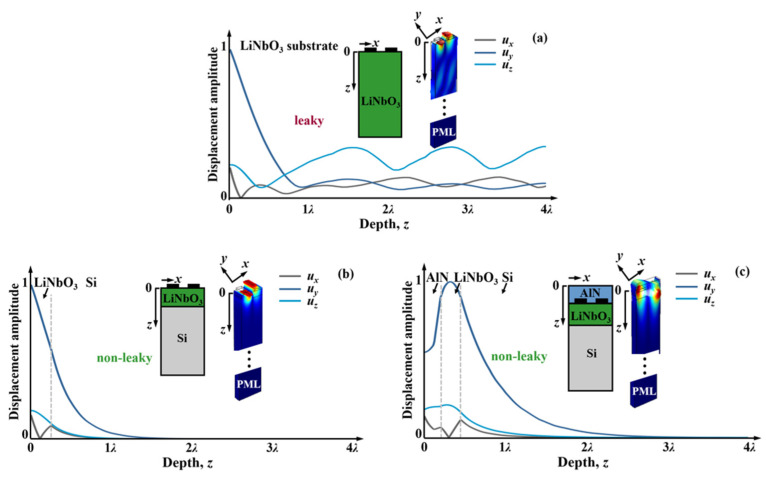
Comparison of simulated displacement amplitudes of (**a**) LiNbO_3_ substrate, (**b**) LiNbO_3_/Si, and (**c**) AlN/ LiNbO_3_/Si for the SH wave. The insets are the mode shapes at antiresonances simulated with periodic structures using FEM simulation (*h*_LiNbO3_/*λ* = 0.25, *h*_AlN_/*λ* = 0.2, *h*_IDT,Al_/*λ* = 0.08 are used in the plots).

**Figure 2 micromachines-13-01029-f002:**
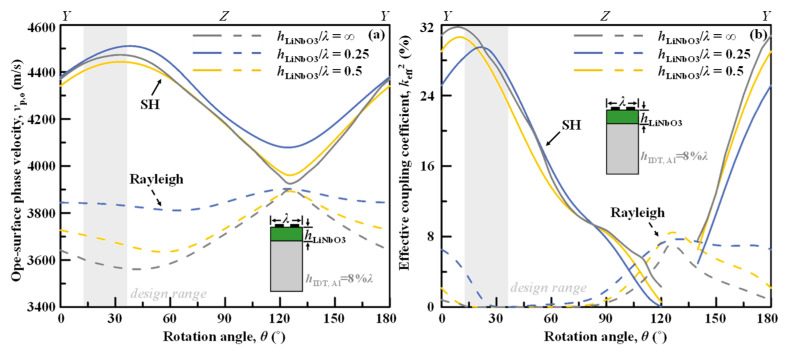
Simulated (**a**) open-surface phase velocities *v*_p,o_ and (**b**) effective coupling coefficient *k*_eff_^2^ for LiNbO_3_/Si across all rotation angles from the *YX*-LiNbO_3_ with varied thicknesses of LiNbO_3_ for the SH main mode and Rayleigh spurious mode.

**Figure 3 micromachines-13-01029-f003:**
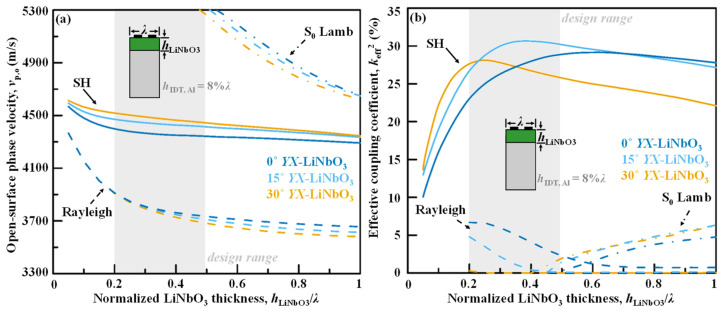
Calculated dispersion curves of (**a**) open-surface phase velocities *v*_p,o_ and (**b**) effective coupling coefficient *k*_eff_^2^ for the LiNbO_3_/Si with varied rotation angle of *YX*-LiNbO_3_.

**Figure 4 micromachines-13-01029-f004:**
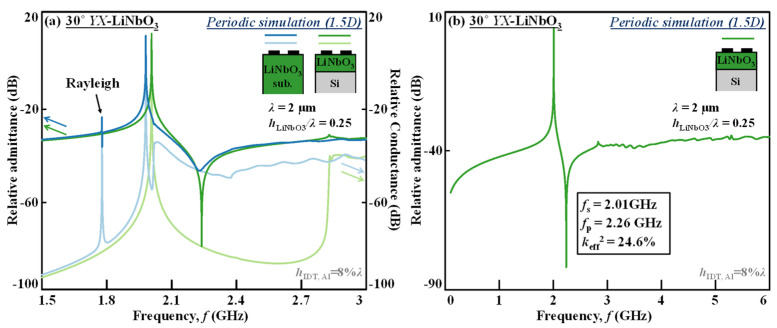
FEM-simulated (**a**) narrow-band response compared to LiNbO_3_ substrate, and (**b**) wide-band response of the SH resonators with periodic structure (1.5D model) based on LiNbO_3_/Si with 30° *YX* LiNbO_3_ and *h*_LiNbO3_*/λ* = 0.25.

**Figure 5 micromachines-13-01029-f005:**
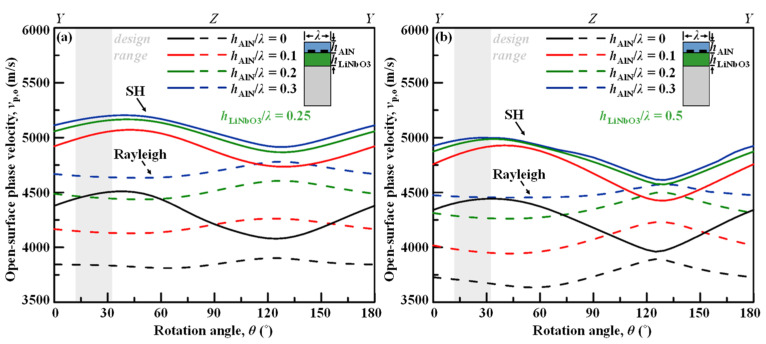
Calculated open phase velocities *v*_p,o_ for varied thicknesses of AlN overlay layer on LiNbO_3_/Si across all rotation angles from the *YX*-LiNbO_3_ with (**a**) *h*_LiNbO3_*/λ* = 0.25, and (**b**) *h*_LiNbO3_*/λ* = 0.5.

**Figure 6 micromachines-13-01029-f006:**
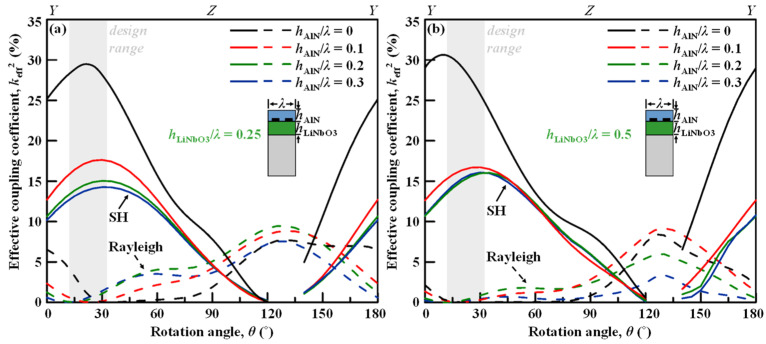
Calculated coupling coefficient *k*_eff_^2^ for varied thicknesses of AlN overlay layer on LiNbO_3_/Si across all rotation angles from the *YX*-LiNbO_3_ with (**a**) *h*_LiNbO3_*/λ* = 0.25, and (**b**) *h*_LiNbO3_*/λ* = 0.5.

**Figure 7 micromachines-13-01029-f007:**
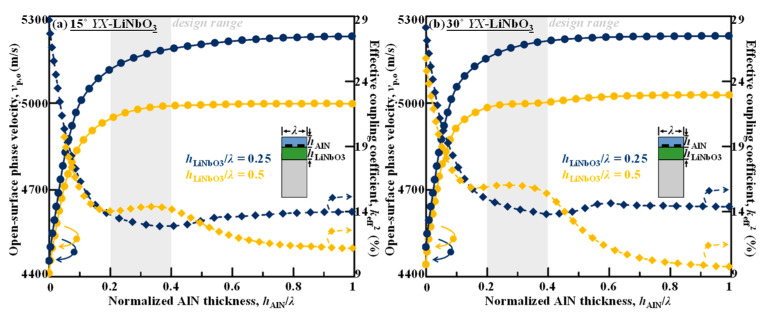
Trade-offs between the open-surface phase velocities *v*_p,o_ and coupling coefficient *k*_eff_^2^ by varying AlN thickness for the SH wave propagating in AlN/ LiNbO_3_/Si based on (**a**) 15° *YX*-LiNbO_3_ and (**b**) 30° *YX*-LiNbO_3_ with *h*_LiNbO3_*/λ* = 0.25 and *h*_LiNbO3_*/λ* = 0.5.

**Figure 8 micromachines-13-01029-f008:**
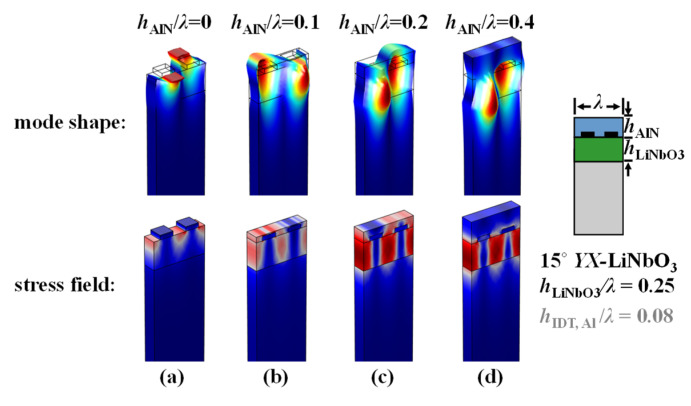
FEM-simulated mode shapes and the first principal stress field of the nonleaky SH mode on the AlN/LiNbO_3_/Si substrate stack with (**a**) *h*_AlN_*/λ* = 0, (**b**) *h*_AlN_*/λ* = 0.1, (**c**) *h*_AlN_*/λ* = 0.2, and (**d**) *h*_AlN_*/λ* = 0.4.

**Figure 9 micromachines-13-01029-f009:**
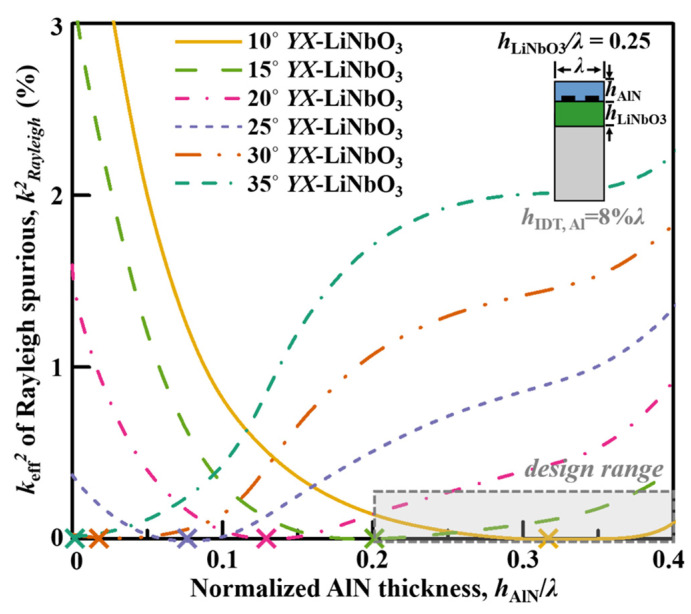
Simulated *k*_eff_^2^ of the Rayleigh spurious mode versus AlN thicknesses for the AlN/ LiNbO_3_/Si with different rotation angles and *h*_LiNbO3_*/λ* = 0.25. The cross labels mark the AlN thicknesses that enables full suppression of the Rayleigh mode.

**Figure 10 micromachines-13-01029-f010:**
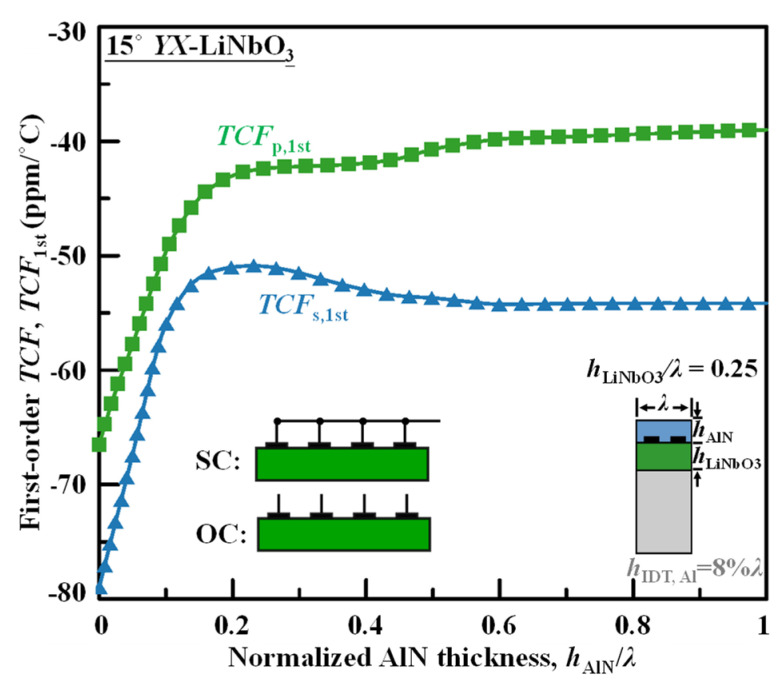
Calculated *TCF*_s,1st_ and *TCF*_p,1st_ by varying AlN thickness for the SH wave propagating in AlN/LiNbO_3_/Si. The inset depicts the short-circuited and open-circuited grating conditions, corresponding to resonance and parallel resonance, respectively.

**Figure 11 micromachines-13-01029-f011:**
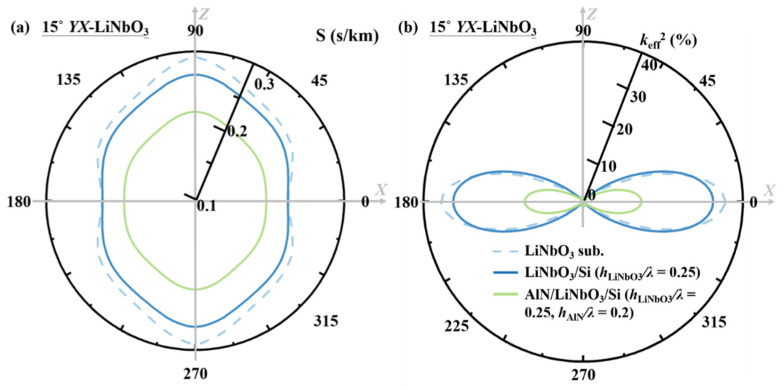
Calculated (**a**) slowness curve; (**b**) effective coupling coefficient *k*_eff_^2^ of the SH wave versus different propagation direction on the 15° *YX*-LiNbO_3_ substrate, LiNbO_3_/Si (*h*_LiNbO3_*/λ* = 0.25), and AlN/ LiNbO_3_/Si (*h*_LiNbO3_*/λ* = 0.25, *h*_AlN_*/λ* = 0.2).

**Figure 12 micromachines-13-01029-f012:**
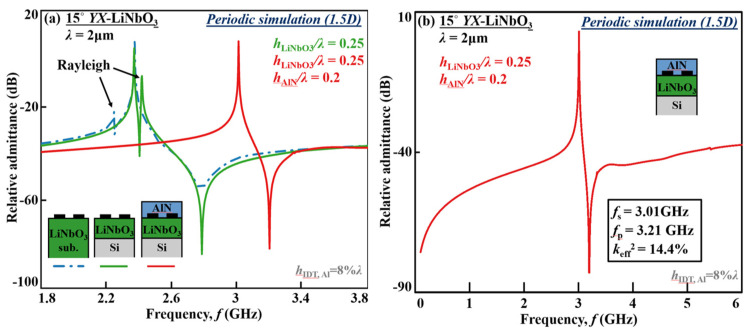
FEM-simulated (**a**) narrow-band response based on AlN/LiNbO_3_/Si (*h*_LiNbO3_*/λ* = 0.25, *h*_AlN_*/λ* = 0.2) compared to LiNbO_3_/Si (*h*_LiNbO3_*/λ* = 0.25) and Si substrate; and (**b**) wide-band response based on AlN/LiNbO_3_/Si (*h*_LiNbO3_*/λ* = 0.25, *h*_AlN_*/λ* = 0.2) for the nonleaky SH resonators with periodic structure (1.5D model).

**Table 1 micromachines-13-01029-t001:** Symbols used in this paper.

Symbol	Definition
*Q*	Quality Factor
*k* _eff_ ^2^	Electromechanical Coupling Coefficient
*λ*	Acoustic Wavelength
*f* _s_	Series-frequency
*u* _x_	*x*-direction displacement
*u* _y_	*y*-direction displacement
*u* _z_	*z*-direction displacement
*h* _LiNbO3_	The thickness of LiNbO_3_
*h* _AlN_	The thickness of AlN
S_0_ mode	The lowest symmetric mode
*v* _p,o_	Open phase velocity
*TCF*	Temperature coefficients of frequency
*TCV*	Temperature coefficients of velocity

**Table 2 micromachines-13-01029-t002:** LiNbO_3_, AlN, Si, and Al material constants used in the simulations [14,15,16].

Parameter	Symbol	LiNbO_3_ [14]	AlN [15]	Si [16]	Al [16]	Units
Stiffness constants	*C* _11_ ^E^	198.39	345	170	111	(GPa)
	*C* _12_ ^E^	54.72	125		59	
	*C* _13_ ^E^	65.13	120		59	
	*C* _14_ ^E^	7.88	–		–	
	*C* _33_ ^E^	227.90	395		111	
	*C* _44_ ^E^	59.65	118		26	
	*C* _66_ ^E^	71.84	110		26	
Density	*ρ*	4628	3260	2329	2700	(kg/m^3^)
Piezoelectric constants	*e* _15_	3.69	−0.48	–	–	(C/m^2^)
	*e* _22_	2.42	–	–	–	
	*e* _31_	0.30	−0.58	–	–	
	*e* _33_	1.77	1.55	–	–	
Dielectric constants	*ε* _11_ ^S^	45.6	8.0	11.7	1	(10^−1^^1^ F/m)
	*ε* _33_ ^S^	26.3	9.5	11.7	1	

**Table 3 micromachines-13-01029-t003:** Temperature coefficients of material constants of LiNbO_3_, AlN, Si, and Al used in the simulations [15,16,21].

Parameter	Symbol	LiNbO_3_ [21]	AlN [15]	Si [16]	Al [15]	Units
Temperature coefficients of stiffness constants (1st order)	*Tc_11_*	−174	−37	−63	−590	(10^–6^ 1/°C)
	*Tc_12_*	−252	−1.8		−80	
	*Tc_13_*	−159	−1.8		−80	
	*Tc_14_*	−214	–		–	
	*Tc_33_*	−153	−65		−590	
	*Tc_44_*	−204	−50		−520	
	*Tc_66_*	−143	−57		−520	
Temperature coefficients of piezoelectric constants	*Te_15_*	147	–	–	–	(10^–6^ 1/°C)
	*Te_22_*	79				
	*Te_31_*	221				
	*Te_33_*	887				
Temperature coefficients of permittivity	*Tε_11_*	323	–	–	–	(10^–6^ 1/°C)
	*Tε_33_*	627				
Thermal-expansion coefficients	*α* _11_	15.4	5.27	2.6	18	(10^–6^ 1/°C)
	*α* _33_	7.5	4.15	2.6	18	

## Data Availability

The data that support the finding of this study are available from the corresponding author upon reasonable request.

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
