# Peer review of "High-Frequency and Spectrum-Clean Shear-Horizontal Acoustic Wave Resonators with AlN Overlay"

_micromachines, 2022, doi:10.3390/mi13071029_

Round 1

Reviewer 1 Report

Dear Authors,

The manuscript addresses a fundamental problem and fits the main scope of the journal. However, there are a few comments and suggestions that should be clearly addressed. Below are some of these major issues:

                     The introduction is not enough. 

                     Many previous works have been cited but I could not see them in the manuscript. 

                     Please correct the x-y axes in Fig. 1

                     FIG is not recommended for MDPI journals, please replay it with Figure.

                     The authors claim that in Fig. 1 “The insets are the mode shapes at anti-resonance simulated with periodic structures using FEM simulation”. I believe this is not a basic characteristic of anti-resonance compared to that presented in Fig. 4.

                     There is an additional ) in table 1.

                     I would recommend adding a table listing all the symbols with brief definitions.

                     Fig. 2 shows a veering phenomenon when the two readings approach each other at a cut angle of 120 degrees. Any clarification about this interesting behavior?

                     After doing this optimization process, can you please provide a map illustrating the main control parameters that could enhance the performance?

                     Have you considered the nonlinearities in this analysis since you are dealing with waves?

                     The main objective of the work is not clear. I think it is highly recommended to describe the main purpose and application.

                     Be careful about MDPI style.

Author Response

Thank you for your comment. We have responsed to the review's comments. Please see the attachment.

Reviewer 2 Report

The simulation results provide clear design information for stacking SAW device. Phase velocity, coupling coefficient, TCF, Q, keff, and fs are covered in the simulation. The draft is well organized and clearly presented. 

Great work.

Author Response

Thank you very much for your kind support . 

Round 2

Reviewer 1 Report

Thanks for the corrections. The paper is in good standing. 

Author Response

Thank you very much for your kind support.